# Curcumin Nanodiscs Improve Solubility and Serve as Radiological Protectants against Ionizing Radiation Exposures in a Cell-Cycle Dependent Manner

**DOI:** 10.3390/nano12203619

**Published:** 2022-10-15

**Authors:** Angela C. Evans, Kelly A. Martin, Manoj Saxena, Sandra Bicher, Elizabeth Wheeler, Emilio J. Cordova, Christopher D. Porada, Graça Almeida-Porada, Takamitsu A. Kato, Paul F. Wilson, Matthew A. Coleman

**Affiliations:** 1Department of Radiation Oncology, University of California Davis, Sacramento, CA 95817, USA; 2Physical and Life Sciences Directorate, Lawrence Livermore National Laboratory, Livermore, CA 94550, USA; 3Institute of Structural and Molecular Biology, Birkbeck College, University of London, London WC1E 7HX, UK; 4Institute of Radiation Medicine, Helmholtz Zentrum München, 85764 Munich, Germany; 5Department of Radiation Oncology, Klinikum Rechts der Isar, Technical University Munich (TUM), 81675 Munich, Germany; 6Engineering Directorate, Lawrence Livermore National Laboratory, Livermore, CA 94550, USA; 7National Institute of Genomic Medicine, Oncogenomic Consortium, Mexico City 14610, Mexico; 8Wake Forest Institute for Regenerative Medicine, Wake Forest University School of Medicine, Winston-Salem, NC 27157, USA; 9Department of Environmental and Radiological Health Sciences, Colorado State University, Fort Collins, CO 80523, USA; 10Earth and Biological Sciences Directorate, Pacific Northwest National Laboratory, Richland, WA 99354, USA

**Keywords:** curcumin, nanodisc, ionizing radiation, radioprotector, radiosensitizer

## Abstract

Curcumin, a natural polyphenol derived from the spice turmeric (*Curcuma longa*), contains antioxidant, anti-inflammatory, and anti-cancer properties. However, curcumin bioavailability is inherently low due to poor water solubility and rapid metabolism. Here, we further refined for use curcumin incorporated into “biomimetic” nanolipoprotein particles (cNLPs) consisting of a phospholipid bilayer surrounded by apolipoprotein A1 and amphipathic polymer scaffolding moieties. Our cNLP formulation improves the water solubility of curcumin over 30-fold and produces nanoparticles with ~350 µg/mL total loading capacity for downstream in vitro and in vivo applications. We found that cNLPs were well tolerated in AG05965/MRC-5 human primary lung fibroblasts compared to cultures treated with curcumin solubilized in DMSO (curDMSO). Pre-treatment with cNLPs of quiescent G0/G1-phase MRC-5 cultures improved cell survival following ^137^Cs gamma ray irradiations, although this finding was reversed in asynchronously cycling log-phase cell cultures. These findings may be useful for establishing cNLPs as a method to improve curcumin bioavailability for administration as a radioprotective and/or radiomitigative agent against ionizing radiation (IR) exposures in non-cycling cells or as a radiosensitizing agent for actively dividing cell populations, such as tumors.

## 1. Introduction

For centuries, turmeric (*Curcuma longa*) has been used in southeast Asia and the Middle East as a curative for minor ailments as well as a food additive, dye colorant, and cosmetic additive [1]. Curcumin, derived from rhizomes of the spice turmeric, is a naturally occurring phytochemical known for its pleotropic health benefits, including antioxidant and anti-inflammatory properties [2]. Curcumin has been studied as a health supplement for a variety of diseases and disorders, including liver fibrosis, diabetes, cardiovascular disease, and cancer [3]. To date, there have been over 100 clinical trials involving more than 6000 participants examining the safety, efficacy, and health benefits of curcumin against numerous disease states [4], including type II diabetes [5], radiation dermatitis [6], or thalassemia [7]. It has been established that curcumin is safe and well tolerated in humans, with acceptable intake up to 12 g/day demonstrating little to no toxicity [8].

The mechanisms of action following curcumin treatment are currently being investigated, including in conjunction with radiotherapy, to act as either a radioprotector in non-cycling cell populations or as a radiosensitizer in cycling cell populations [2,3,4,9]. It has been shown that curcumin modulates a wide variety of cellular signaling pathways that affect multiple downstream functions including cell proliferation, apoptosis, cell cycle arrest, and inflammatory processes [3,4]. For example, curcumin has been shown to increase cell cycle arrest and disrupt vascular endothelial growth factor (VEGF), epidermal growth factor receptor (EGFR), and cyclooxygenase-2 (COX-2) pathways in numerous cancer cell lines [3]. Curcumin is also a potent reactive oxygen/nitrogen species (ROS/RNS) scavenger, and it has been proposed that curcumin alters arachidonic acid (AA) metabolism, potentially as an upstream regulator of inflammatory processes via cyclooxygenase (COX), lipoxygenase (LOX), and tumor necrosis factor (TNF) inhibition [10]. Curcumin has also been shown to decrease ROS/RNS production by decreasing inducible nitric oxide synthase (iNOS), increasing glutathione peroxidase, as well as increasing the transcription of antioxidant gene response elements through the Nrf2-Keap1 pathway [10].

In addition to generating direct DNA damage, ionizing radiation (IR) exposures increase ROS/RNS production, which can lead to increased oxidative DNA damage in cells and result in the activation of downstream DNA damage response (DDR) pathways. By mitigating ROS/RNS production, it has been demonstrated that natural antioxidants, such as curcumin, may be useful as radiation medical countermeasures (MCM) [11]. Curcumin treatments have been shown to protect against lipid peroxidation and improve antioxidant potential following up to 4 Gy ^60^Co gamma ray exposures in human lymphocytes [12]. Curcumin treatments have also improved wound healing response times in mice, following exposure to fractionated ^60^Co gamma rays [13], and oral administration of curcumin in mice showed decreased levels of micronucleated polychromatic erythrocytes following 1.15 Gy of total body ^60^Co gamma irradiation [14]. Interestingly, a study using human peripheral blood lymphocytes (PBLs) showed reduced levels of prematurely condensed chromosome (PCC) fragments in curcumin-treated quiescent cultures exposed to up to 6 Gy ^60^Co gamma irradiation; however, curcumin treatments resulted in increased levels of chromatid-type chromosomal aberrations in mitogen-stimulated PBLs using the G2 chromosomal radiosensitivity assay [9]. These studies collectively demonstrate a radioprotective effect of curcumin when cells are not actively cycling due to its ROS/RNS scavenging ability; however, in cycling cell populations (e.g., tumor cells which are difficult to maintain in a quiescent state), curcumin has a radiosensitizing effect due to perturbations in post-irradiation DDR pathway activation.

Although these studies suggest curcumin may be a promising phytochemical candidate for a radiation MCM in quiescent cell populations and tissues, its inherently low water solubility and poor bioavailability results in limited absorption, rapid metabolism, and fast excretion [15]. Several formulations have been proposed to circumvent the problem of low curcumin bioavailability, including curcumin-adjuvant formulations, nanoparticles, or liposomes [15,16,17]. Liposomal curcumin formulations have been shown to enhance curcumin bioavailability in vitro and in vivo via increasing apoptosis and toxicity to colon, lung, prostate, and cervical cancer cell lines, as well as to decrease tumor growth in mouse xenograft models [18,19]. Polymeric curcumin nano-formulations also improve curcumin water solubility and are well tolerated in vivo [20]. In this study, we describe the formulation and characterization of curcumin nanolipoprotein particles (cNLPs) using typical cell membrane-derived components (phospholipids, apolipoproteins) as delivery vehicles to increase curcumin solubility and bioavailability for radiobiological studies.

NLPs are nanodiscs composed of phospholipids and scaffolding proteins that mimic high-density lipoprotein (HDL) particles within the human body. They are readily customizable to include a variety of lipids, scaffolding proteins, as well as polymeric or adjuvant formulations, and are virtually non-toxic in vitro and in vivo [21]. In addition, NLPs also act as efficient biomimetic transporters of hydrophobic cargo into cells [21], serving as biodelivery platforms for otherwise unstable or insoluble moieties including self-amplifying mRNA replicons [22] and membrane proteins [23,24]. It has previously been shown that curcumin incorporation into stable nanodiscs surrounded by either apolipoprotein ApoA1 or ApoE increases the efficiency of curcumin delivery and apoptosis induction in hepatoma, lymphoma, and glioblastoma cell lines [25,26,27]. However, these studies did not examine the effects of curcumin nanodiscs on normal, non-transformed human cells, nor evaluate its impact on cells treated in different stages of the cell cycle after ionizing radiation. Here, we introduce the production of curcumin-loaded ApoA1 NLPs (cNLPs) by first mixing curcumin with 1,2-dimyristoyl-*sn*-glycero-3-phosphocholine (DMPC) lipid and amphipathic telodendrimer polymers, followed by the addition of ApoA1 scaffolding proteins, which represents a novel formulation. We also characterized curcumin bioavailability in vitro as a potentially less toxic alternative to DMSO-solubilized curcumin. Moreover, we examined the effects of these cNLP formulations on IR-induced DNA damage and cell survival in vitro.

The current study examines the effects of cNLP treatments compared to DMSO-solubilized curcumin using low-passage, apparently normal, primary lung fibroblasts (AG05965/MRC-5) via DNA double-strand break (DSB)-associated foci induction and repair kinetic analyses, as well as single cell colony formation assays following exposure to ^137^Cs gamma rays. The goals of this study were to demonstrate the feasibility of producing cNLPs, characterize any toxicity they may have in vitro, and test whether cNLPs modulate post-irradiation cell survival in both quiescent G0/G1-phase and asynchronous log-phase cultures of MRC-5 fibroblasts. Such fundamental work will have broader utility in radiotherapy and for treating occupational exposures to radiation (e.g., nuclear industry workers, radiological technologists, astronauts).

## 2. Materials and Methods

### 2.1. Formulating Curcumin-Telodiscs (Cur-Telodiscs)

The first step to cNLP formation is the generation of curcumin telodendrimer-lipid discs (cur-telodiscs). To formulate cur-telodiscs, 60 mg/mL curcumin in DMSO, 20 mg/mL of 1,2-dimyristoyl-*sn*-glycero-3-phosphocholine (DMPC) lipid (Avanti Polar Lipids) in dH_2_O, and 20 mg/mL telodendrimer PEG5k-CA8 polymers in dH_2_O were prepared as stock solutions. Reconstituted DMPC lipid [20 mg/mL] was sonicated in a horn sonicator for ~25–45 min until optical translucence was achieved. DMPC lipid was then mixed with 20 mg/mL telodendrimers in a 10:1 vol/vol (DMPC:telodendrimer) ratio. Solubilized curcumin [60 mg/mL in DMSO] was then added to the lipid–telodendrimer mixture at a 9:11 vol/vol ratio (curcumin:DMPC/telodendrimer mixture) yielding final cur-telodisc concentrations of 27 mg/mL curcumin, 10 mg/mL DMPC, and 1 mg/mL telodendrimer. The resulting cur-telodiscs were then sonicated for 30–45 min in a horn sonicator and spun down in microcentrifuge tubes at 10,000 rpm for 1 min to remove any unincorporated curcumin.

### 2.2. Formulating Curcumin Nanolipoprotein Particles (cNLPs)

To formulate cNLPs, 500 µL of soluble cur-telodisc solution was combined with purified Δ49ApolipoproteinA1 (ApoA1) in 1× PBS to a final ApoA1 concentration of 2 mg/mL and final volume of 1 mL. The resulting curcumin nanoparticles were sonicated for 30 min in a bath sonicator and dialyzed overnight (membrane cutoff 3.5 kDa) against PBS pH 7.4 at room temperature. After dialysis, 250 µL of freshly sonicated [20 mg/mL] DMPC lipid was added per ml of dialyzed curcumin mixture, re-sonicated for 45 more minutes in a bath sonicator, and centrifuged at 10,000 rpm for 1 min to remove any unincorporated curcumin. The resulting supernatant contained soluble curcumin nanolipoprotein particles (cNLPs).

### 2.3. UV-Vis Spectroscopy

UV-Vis spectroscopy was utilized to determine total curcumin incorporation levels in the cNLPs based on previous studies using ApoA1 and E4 nanodiscs and curcumin-amyloid aggregates [25,26,27]. Curcumin absorbance at 430-nm was read using the NanoDrop^TM^ One^C^ spectrophotometer (Thermo Fisher Scientific, Waltham, MA, USA). To establish a standard curve, 430-nm absorbance readings were collected for two-fold serial dilutions of curcumin solubilized in DMSO at 0.25 mg/mL, 0.125 mg/mL, 0.06 mg/mL, 0.03 mg/mL, and 0.015 mg/mL. The resulting standard curve equation was calculated using Microsoft Excel along with resulting R^2^ values. For our experiments, the R^2^ values were consistently ≥0.99. New standard curves were created with fresh DMSO-solubilized curcumin for comparison to each new cNLP prep. Absorbance readings of new cNLP preps at 430-nm were then compared to these standard curves to give final concentrations of curcumin in our cNLP samples.

### 2.4. Size Exclusion Chromatography (SEC), Dynamic Light Scattering (DLS), and UV Transluminescence

Size exclusion chromatography (SEC) was used to demonstrate curcumin incorporation into the cNLPs. cNLPs were diluted ~1:50 in PBS and loaded into a SEC column (Superose 6 Increase, GE Healthcare, Chicago, IL, USA) at 50 µL injection/sample and run at a flow rate of 0.6 mL/min in PBS buffer. Curcumin absorbance (430-nm) along with protein absorbance (280-nm) was used to demonstrate curcumin incorporation into the cNLPs. Total run times for each sample did not exceed 6 min. Dynamic light scattering (DLS) was applied to determine the size of the curcumin-loaded nanoparticles. Here, 1 µL of cNLP solution was diluted in 99 µL PBS in a cuvette. The cuvette was then placed into the DLS (Zetasizer Nano ZSP, Malvern Panalytical, Malvern, UK) and the average particle size and polydispersity index were determined using Zetasizer Nano ZSP software version 7.12. UV Transluminescence was applied to visualize fluorescence of curcumin dissolved in water or DMSO compared to curcumin in cur-telodisc form. 100 µL of each solution was aliquoted and imaged for fluorescence intensity under a UV lamp (365-nm).

### 2.5. Cell Culture

AG05965/MRC-5 apparently normal primary human lung fibroblasts were obtained from the Coriell Cell Repositories (CCR) and maintained in αMEM with 1X GlutaMAX media (GIBCO/Invitrogen, Waltham, MA, USA). These cells were chosen due to their prior use in previously published in vitro radiobiology experiments. Media was supplemented with 15% FBS (Sigma, Burlington, MA, USA), 0.8% PenStrep (GIBCO/Invitrogen), 0.8% MEM Vitamins (GIBCO/Invitrogen), 0.8% MEM (Essential) Amino Acids Solution (GIBCO/Invitrogen), 0.8% MEM Non-Essential Amino Acids Solution (GIBCO/Invitrogen), and 4 mM NaOH. Cell culture media was filtered through a 0.2-micron PES filter before use. Cells were cultured in a 37 °C incubator supplied with 95% air/5% CO_2_ mixture. All experiments were conducted with low-passage cultures (≤passage 20).

### 2.6. Mitotic Index Assay

Asynchronous log-phase AG05965/MRC-5 fibroblasts were treated with 10 µg/mL [27 µM] cNLPs, 10 µg/mL [27 µM] curDMSO, 10 µg/mL empty NLP, or 0.1% DMSO for 18 h after which cells were trypsinized, fixed in 70% ethanol and stored at −20 °C overnight. Cells were treated with 0.2% Triton X-100 solution in PBS for 10 min at RT. After blocking for 30 min in 10% goat serum in PBS, cells were treated with anti-histone H3 phospho-serine 10 rabbit polyclonal antibody (1:500 dilution; Abcam #Ab5176) for 1 h at 37 °C, washed with PBS, and treated with Alexa Fluor™ 546-conjugated highly cross-adsorbed anti-rabbit IgG (H + L) secondary antibody (1:500 dilution, Thermo Fisher #A11035) for 1 h at 37 °C. Phospho-histone H3-positive cells were analyzed and quantitated with a Guava PCA flow cytometer (Sigma) with a minimum of 10,000 cells analyzed per sample treatment.

### 2.7. Cell Proliferation (MTS) Assay

AG05965/MRC-5 cells were seeded at 10,000 cells/well in a 96 well plate. After ~24–48 h of growth, the cells were washed with PBS then treated with cNLPs [0, 20, 40, 60, 80, or 100 µM curcumin], curDMSO [0, 20, 40, 60, 80, or 100 µM curcumin], or empty NLPs [added the same volumes as cNLP group] for 18 h. The maximum DMSO concentration used in the curDMSO group was 0.4%. Following 18-h curDMSO/cNLP/empty NLP treatments, 20 µL of CellTiter 96^®^ AQueous One Solution Reagent (Promega) was added to each well containing the samples in 100 µL culture medium. Cells were then incubated at 37 °C for 4 h. Absorbance was read at 490-nm using a 96-well plate reader with 650-nm background absorbance subtracted from all wells prior to normalization. Normalized values were calculated by dividing the mean 490-nm absorbances of empty NLP, cNLP, or curDMSO treated wells by the average 490-nm absorbances from cells treated only with cell culture media (Absorbance_490_ treated cells/Absorbance_490_ media alone cells). Two independent biological replicates were combined to determine statistical significance using two-way ANOVA (Graphpad Prism, Version 9.1.2).

### 2.8. DNA Damage Immunohistochemistry

AG05965/MRC-5 fibroblasts were grown to density-inhibited confluency in Nalge-Nunc flaskettes (Thermo Fisher Scientific). Cells were then treated for 18 h with 10 µg/mL [27 µM] cNLPs, 10 µg/mL [27µM] curDMSO, or 0.1% DMSO alone prior to 50 cGy ^137^Cs gamma ray or sham (0 cGy) irradiations. Flaskettes were then returned to the incubator and fixed at 15, 120, 360 and 1440 min post-irradiation by aspirating medium, rinsing gently twice with Dulbecco’s PBS (DPBS +Ca^2+/^Mg^2+^), and fixed for 15 min with freshly prepared 4% paraformaldehyde (PFA) in PBS. Following fixation, PFA was aspirated and flaskettes were rinsed three times for 5 min with DPBS on rocker and then filled with DPBS, capped, and resealed with parafilm and stored in a 4 °C refrigerator until immunostaining.

Cells were co-stained for γ-H2AX pS139 and 53BP1 DSB-associated foci via immunocytochemistry, as previously described [28]. Briefly, chambers were rinsed twice with room temperature (RT) DPBS for 5 min each. Cells were permeabilized with ice-cold PBS with 0.5% Triton X-100 and placed on rocker on ice tray for 10 min, after which, the permeabilization solution was aspirated and flaskettes rinsed twice with RT DPBS for 5 min each. Samples were treated with Image-iT™ FX Signal Enhancer solution (Thermo Fisher), and incubated at room temperature for 30 min, then, chambers were rinsed twice with RT DPBS for 5 min each. Samples were blocked with 2% goat serum, 2% FBS, 1% BSA, 0.1% Triton X-100, and 0.05% Tween-20 in PBS at RT for 30–60 min. After blocking, cells were rinsed quickly with RT DPBS and flaskette chambers and gaskets carefully removed. Primary antibody solutions of 1:250 mouse anti-phosphoH2AX pS139 clone JBW301 (Millipore #05-636) and 1:250 rabbit anti-53BP1 (Novus #NB100-304) in PBS with 1% BSA were loaded on slides. Slides were overlayed with coverslip and incubated horizontally in a 37 °C humid chamber for 30–60 min. Slides were rinsed three times in RT DPBS for 5 min each in dark Coplin jars on rocker. Secondary antibody solutions consisting of 1:500 Alexa 488™-conjugated goat anti-mouse IgG F(ab’)2 fragments (Thermo Fisher #A-11017) and 1:500 Molecular Probes Alexa 594™-conjugated goat anti-rabbit IgG F(ab’)2 fragments (Thermo Fisher #A-11072) prepared in PBS were then mounted on slide with overlaying coverslip and incubated horizontally in a dark 37 °C humid chamber for 30–60 min. Slides were then rinsed twice with RT DPBS for 5 min each in dark Coplin jars on rocker, followed by one final rinse for 10 min. Slides were then incubated in freshly prepared RT 3.7% paraformaldehyde in PBS for 10 min, followed by two 5-min rinses with RT DPBS. After the final rinse, excess PBS was removed, and slides were treated with RT Molecular Probes ProLong Gold with 0.2 µg/mL DAPI (Thermo Fisher) and overlayed with a 24 mm × 50 mm coverslip. Slides were cured overnight at RT in a light-tight slide box, then transferred to the refrigerator the following day and stored at 4 °C until imaging. Slides were imaged with a Carl Zeiss microscope equipped with a 63× objective using Zen Software. Over 250 cells were counted per experimental time point, with at least two biological replicates performed.

### 2.9. Survival Curves

To assess the effects of cNLPs on post-irradiation cell survival, quiescent G0/G1-phase AG05965/MRC-5 fibroblasts were treated for 18 h with 10 µg/mL [27 µM] cNLPs, 10 µg/mL [27 µM] curDMSO, 10 µg/mL empty NLPs, or 0.1% DMSO alone. After 18-h treatments, cells were trypsinized and counted via Countess II FL (Invitrogen) and aliquoted into respective tubes for irradiation. Appropriate numbers of cells were aliquoted to yield ~50–100 viable colonies per flask after 14 days of undisturbed growth at 37 °C. Immediately following irradiation, cells were plated in triplicate T-25 flasks and placed in 37 °C incubator with 95% air/5% CO_2_ mixture with a fresh media change on day 7. At 14 days post-irradiation, the medium in the flasks was aspirated, the flasks were twice rinsed gently with DPBS and fixed/stained with 3–4 mL of fixative/staining solution (50% *v*/*v* Kopykake blue dye, 40% *v*/*v* 95% EtOH, and 10% *v*/*v* glacial acetic acid). Flasks were stained on a rocker for 30–60 min, after which, they were rinsed gently twice with dH_2_O. Colonies with ≥50 cells were scored as survivors using a standard light microscope.

### 2.10. Statistical Analysis

A *p*-value of 0.05 was used as a cutoff to determine statistical significance. *p*-values are labeled as <0.05 (*), <0.01 (**), <0.001 (***), and <0.0001 (****) throughout the manuscript. Brown–Forsythe ANOVA was utilized for the mitotic index assay. Two-way ANOVA was used for the MTS assay. For foci counts, one-way ANOVA was applied amongst sham-irradiated control groups at 15 min and 24-h time points and two-way ANOVA was used for comparing IR-induced (background-corrected) foci/cell levels. Statistical analyses and graphs were generated using Graphpad Prism Version 9.1.2 software.

## 3. Results

### 3.1. Nanolipoprotein Particles Support Curcumin Loading

Curcumin was successfully incorporated into nanodiscs using a combination of 1,2-dimyristoyl-*sn*-glycero-3-phosphocholine (DMPC), amphiphilic telodendrimer polymer [29], and Δ49 apolipoprotein A1 (ApoA1) (Figure 1). Size exclusion chromatography (SEC) illustrates a mixture of nanodiscs containing curcumin surrounded by either telodendrimer polymers or apolipoprotein-dominant cNLP products (Figure 2A). Nanodiscs containing curcumin, DMPC, telodendrimer, and apolipoprotein ApoA1 scaffold are all components of larger nanoparticles that elute from the SEC column at around 2 min. In contrast, smaller nanodiscs were formed and eluted with a distinct peak at about 3.6 min. Both species demonstrate overlapping 280-nm and 430-nm absorbances, indicating incorporated curcumin within the NLP nanodiscs (Figure 2A). The resulting mixture of curcumin nanodiscs (mixtures of smaller curcumin-apolipoprotein A1 and larger curcumin-telodendrimer-apolipoprotein A1 discs) were applied to all downstream assays and were collectively termed “cNLPs”. Dynamic light scattering (DLS) estimated an average size of 43 nm for our heterogeneous cNLP formulation (standard deviation of 11 nm) with a polydispersity index of 0.232 (Figure 2B). Within the nanodisc environment, we also saw increased solubility over curcumin solubilized in water and a similar solubility to curcumin solubilized in DMSO (curDMSO) (Figure 2C). Contrary to the yellow fluorescence observed under UV light for curDMSO solutions, our curcumin nanodiscs demonstrate a slight blue shift in fluorescence, indicating incorporation of the curcumin into the lipid nanodisc environment (Figure 2C), as has been described previously [30]. Across three biological replicates, our cNLP formulation yielded an average of 348 µg/mL incorporated curcumin.

### 3.2. Normal Human Fibroblasts Tolerate cNLPs Better than DMSO-Solubilized Curcumin

Low-passage AG05965/MRC-5 primary human fibroblasts were treated with increasing concentrations of curcumin solubilized in either nanodiscs (cNLPs) or DMSO (curDMSO), along with empty NLPs (no curcumin) as a control. Treatment with curDMSO decreased cell viability more so than cNLPs at higher curcumin concentrations of 40, 60, 80, and 100 µM (Figure 3). Statistically significant differences (*p* < 0.05) were observed for the 60 µM and 100 µM concentrations. At both 40 µM and 80 µM curcumin concentrations, cells trended towards higher viability in the cNLP groups, but these differences were not statistically significant (*p* = 0.1). Cell viability was similar for the 20 µM cNLP and curDMSO groups, and interestingly, cNLP treatments at this concentration resulted in higher cell viability compared to empty NLP controls (Figure 3). We also found that cNLP and curDMSO treated AG05965/MRC-5 cultures demonstrated higher mitotic indices compared to empty NLP or DMSO vehicle controls, although these differences were not statistically significant (Appendix A).

### 3.3. cNLP Pre-Treatment Alters Foci Persistence Following Gamma Irradiation

With curcumin demonstrating antioxidant ROS/RNS-scavenging properties, we next determined if cNLPs alter DNA DSB foci induction and 24 h repair kinetics following ^137^Cs gamma ray exposures. Background levels of γH2AX pS139/53BP1 foci in sham-irradiated samples fixed at 15 min and 24 h were not statistically different and were averaged for each treatment (cNLP, curDMSO, or DMSO alone) and background-subtracted from total foci counts in irradiated cells to determine levels of IR-induced foci/cell. The numbers of IR-induced foci/cell in cNLP, curDMSO, and DMSO-treated groups exposed to 50 cGy ^137^Cs rays were not significantly different from one another at 15 min or 24 h post-irradiation (Figure 4). For the 2-h time point, cNLP-treated cells displayed increased levels of foci when compared to DMSO controls (mean induced foci/cell 4.84 and 3.77, respectively, *p* = 0.007), whereas curDMSO foci counts showed no significant differences when compared to DMSO controls. Interestingly, cNLP-treated cells were the only group that did not have significant differences in induced foci counts between the 15 min and 2-h time points (4.72 and 4.84/cell, respectively; Appendix A). IR-induced cNLP foci counts at 2 h were not significantly different than those observed for curDMSO treatments (4.84 vs. 4.16 foci/cell, respectively). In addition, both curDMSO and cNLP groups displayed increased foci counts (mean induced foci/cell 2.2 and 2.5, respectively) compared to DMSO alone (mean induced foci/cell 0.94) at 6 h post-exposure (Figure 4). Within each treatment group, the numbers of foci gradually decreased with time, returning to sham-irradiated control levels by 24 h post-irradiation (Appendix A–C).

### 3.4. cNLP Effects Following ^137^Cs Irradiation Cell Survival Are Cell-Cycle Dependent

We next sought to investigate whether cNLPs modulate post-irradiation cell survival using single-cell colony formation assays. In confluent G0/G1 AG05965/MRC-5 cultures pre-treated for 18 h and then exposed to 0–6 Gy of ^137^Cs gamma rays, both 27 µM curDMSO and cNLP treatments resulted in higher relative cell survival compared to empty NLP or DMSO vehicle controls (Figure 5A,B). As shown, the cNLP showed increased cell survival compared to empty NLP as well as curcumin solubilized in DMSO. Given previous reports that curcumin effects may be cell-cycle dependent [9], we next sought to investigate effects of curcumin pre-treatment in asynchronous log-phase cultures prior to irradiation. Asynchronous AG05965/MRC-5 cultures were pre-treated for 18 h with cNLP, curDMSO, or empty NLP/DMSO vehicle and exposed to 0–6 Gy of ^137^Cs gamma rays. The mean relative surviving fraction in 27 µM cNLP-treated groups was less than corresponding empty NLP controls across all IR doses (Figure 5C). Similarly, all curDMSO treated groups had lower mean relative surviving fractions than DMSO control groups across all IR doses (Figure 5D).

## 4. Discussion

We demonstrate that NLPs robustly solubilize curcumin to form cNLPs in a lipidic nano-environment, as evidenced by absorbance shift, size exclusion chromatography (SEC), and dynamic light scattering (DLS)-based measurements. We also observe significantly fewer toxic effects of cNLP administration on MRC-5 cellular viability over DMSO-solubilized curcumin (curDMSO). Furthermore, we reproducibly show that curcumin effects on post-irradiation cell survival are largely cell-cycle dependent, as has been suggested previously [9]. This study demonstrates the utility of cNLPs as a potential radioprotective agent/MCM in non-dividing cells and tissues. This is also the first study on the application of ApoA1-based nanodiscs as a delivery tool for radiation protection.

Curcumin exhibits numerous health benefits, including antioxidant, anti-inflammatory, and free-radical scavenging properties, that act on a variety of cellular mechanisms, such as inhibition of NF-kB, COX-2, and TNF [10,31]. It is well accepted that curcumin modulates a variety of pathways and transcription factors and can act as both a chemosensitizer and radiosensitizer in a multitude of cancer cell lines and xenograft mouse models [31,32,33,34]. Recent studies have highlighted potential mechanisms by which curcumin has altered cancer cell death, including activation of tumor-suppressor genes Tp53 and PTEN while suppressing P13K and mTOR pathways [35,36,37]. Curcumin has also been explored as a modulator of cancer cell apoptosis by suppressing miR-21 [38,39] and miR-19 [40], as well as an inducer of autophagy [41] or senescence [42] in numerous cancer cell lines. Furthermore, curcumin may synergize with other anti-cancer agents, including cisplatin or docetaxel, as well as radiation treatment [43,44,45].

Although curcumin increases cancer cell killing through a variety of mechanisms, it has also been shown that curcumin may, in contrast, provide both chemotherapeutic and radiological protection in normal tissues [32]. Srinivasan et al. demonstrated that curcumin pre-treatment on human peripheral blood lymphocytes was able to protect against micronuclei and dicentric chromosomal formation following up to 4 Gy gamma irradiation exposures [12]. In addition, Abraham et al. illustrated that curcumin treatment 2 h before gamma irradiation protected against chromosomal damage in mice [14]. Additional in vivo studies have shown that mice injected with curcumin either 5 days before or 5 days after a single 50 Gy exposure to the hind leg improved both acute and chronic skin toxicity [46], and curcumin also decreased associated nephrotoxicity and cardiotoxicity following cisplatin or doxorubicin treatments in rats [32]. Furthermore, although curcumin is toxic to cancer stem cells, it demonstrates little to no toxicity to normal stem cells [47]. Potential mechanisms at play distinguishing the toxicity of curcumin on cancer cells versus normal cells have suggested that cancer cells preferentially take up much more curcumin than normal cell counterparts [48], which may enhance cytotoxicity. It has also been demonstrated that this difference in radiation protection capability may be cell-cycle dependent [9].

Although curcumin acts as a pleiotropic polyphenol studied for its ability to protect against a variety of diseases and inflammatory disorders, its bioavailability is limited due to its poor water solubility. Curcumin lipid formulations therefore remain promising to enhance curcumin bioavailability and therapeutic efficacy. Lipid nanoparticles have been shown to increase curcumin bioavailability through in vitro cell cultures and in vivo animal models [21,49,50]. Curcumin nano-formulations have also been explored in the clinic for treating inflammatory diseases. For example, curcumin nano-micelles tested in patients with multiple sclerosis (MS) demonstrated fluctuating miRNA expression levels in peripheral blood lymphocytes following 6 months of treatment, suggesting that curcumin nanodiscs may be immunomodulatory in MS patients [51]. Another clinical trial showed increased survival in amyotrophic lateral sclerosis patients treated with 80 mg/day nanocurcumin for 1 year [52].

In the current study, formulation and characterization of heterogeneous curcumin-loaded nanodiscs (cNLPs) surrounded by a telodendrimer and/or apolipoprotein scaffold were explored as a novel formulation for the potential radiological protection against DNA damage or cellular survival following ^137^Cs exposures. Here, we demonstrated that our cNLPs improve curcumin water solubility, are less toxic than curDMSO, and can be readily taken up in a normal human fibroblast cell line (Figure 2 and Figure 3). We also showed that cNLPs alter the foci persistence of DNA DSBs (Figure 4). Interestingly, in our cell survival experiments, we found that cNLPs act as radiation protectors in AG05965/MRC-5 fibroblasts when irradiated in quiescent G0/G1 cultures across 1–6 Gy doses, but this radioprotective effect was reversed in asynchronous cell cultures, where curcumin acted like a radiation-sensitizing agent (Figure 5). Taken together, these data illustrate that the radioprotective effect of curcumin is cell-cycle dependent, even amongst normal human cell cultures.

Ghosh et al. have previously demonstrated enhanced bioavailability of curcumin nanodiscs over free curcumin (curDMSO) in vitro [25,26]. In these studies, curcumin nanodiscs with ApoA1 scaffold protein showed increased apoptosis in human hepatoma and lymphoma cell lines over curcumin alone controls [25,26], and ApoE curcumin nanodiscs demonstrated increased apoptosis and bioavailability in glioblastoma cells [25]. Compared to these reports, we similarly incorporated curcumin into a nanodisc using a combination of sonication, centrifugation, and dialysis techniques [26]. Here, our final cNLP formulation has similar overall curcumin incorporation (~350 µg/mL) and our nanodiscs are also of comparable average size (~43 nm vs. 48 nm) [26]. However, our methodology of formulating cNLPs differed, as we added telodendrimer polymers to help solubilize curcumin within lipids. Most importantly, we tested our cNLPs on primary human AG05965/MRC-5 fibroblasts, in contrast to previously published data derived from cancer cell lines, demonstrating that our cNLPs enhance the viability and offer some protection against irradiation in normal cells. Our findings also suggest that the radioprotection induced through cNLP pre-treatment may be dependent on cell-cycle, with quiescent cultures demonstrating radioprotection and asynchronous cultures showing radiosensitization.

It is well known that the cellular DNA damage response (DDR) is activated following an ionizing radiation exposure [53]. Phosphorylation of histone H2AX, forming gamma H2AX and inducing subsequent foci, is a well-accepted biomarker of DNA DSBs following numerous qualities of IR exposures [28,54]. Here, we demonstrated co-localized immunofluorescent staining of gamma H2AX and 53BP1, a mediator protein also recruited to the site of DNA damage following genotoxic insult [55]. As may be expected, our co-localized immunostained AG05965/MRC-5 fibroblasts do not detect differences in induced foci formation amongst curcumin treated and DMSO controls 15 min after irradiation, suggesting that curcumin does not affect the induction of DNA DSB following 50 cGy of ^137^Cs exposures. Interestingly, we detected persistent foci formation in cNLPs at 2 h post-exposure compared to DMSO alone controls, however, this was not significantly different than the curDMSO groups. We also saw slightly increased foci formation between both cNLP and curDMSO at the 6-h time point compared to cells that did not receive curcumin treatment (Figure 4). This suggests that curcumin may delay the timing of DNA repair to complete following IR exposure. A previous study using HeLa cells has shown that curcumin treatment suppresses DNA damage repair processes, including non-homologous end joining (NHEJ), homologous recombination (HR), and DNA damage checkpoints [56]. However, this has not yet been confirmed in normal cells nor tested if this DNA repair effect is cell-cycle dependent. By 24 h after irradiation, no significant differences with induced foci/cell were detected between cNLP, curDMSO, or DMSO alone treated groups and all had returned to baseline levels (Figure 4).

In our current study, we see that 10 µg/mL [27 µM] cNLPs protect against normal human AG05965/MRC-5 fibroblast cell death following 1–6 Gy of ^137^Cs exposures, with the greatest enhanced mean overall survival over empty NLP controls at 4 and 6 Gy (Figure 5A). A previous report by Lee et al. also investigated curcumin effects on cell survival in a mouse lung carcinoma cell line and showed that 10–25 µM curcumin also affected cell survival, with curcumin enhancing radiation sensitivity following up to 6 Gy gamma exposures [57]. Here, we demonstrate that in a synchronous, non-actively dividing normal fibroblast cell line, curcumin similarly alters radiation response, but in favor of cell survival, enhancing radiation protection over empty NLPs or curDMSO controls (Figure 5). Although we determined that this protection was not due to decreases in DSB induction, potential mechanisms at play here may be the antioxidant nature of curcumin to scavenge free radicals or increase antioxidant production following radiation exposures [10,57]. Moreover, we demonstrated that this radiation protection effect of curcumin was cell-cycle dependent, as the same cNLP and curDMSO formulations tested on asynchronous normal fibroblast cell cultures led to radiation sensitization across all doses tested (Figure 5C,D).

In this study, although average relative survival of cNLP groups outperformed curDMSO or empty NLP controls in non-actively dividing cultures, we do not see significant differences in the ability of cNLPs to protect against cell survival, as each dose had large variability (Figure 5). This may be due to the concentration of cNLP used in our experiments (10 µg/mL [27 µM]). It has been previously shown that free curcumin is toxic to cancer cells ≥1 µM [58] and we have similarly demonstrated toxicity in K562 and HL-60 human leukemia cell lines treated with 15–30 µM curcumin [59]. However, since normal cells are more resistant to curcumin than cancer cells [47,48], presumably due to cell-cycling differences, it is therefore possible that higher curcumin concentrations (above 30 µM) may be beneficial to cell survival in quiescent cultures following IR exposures. Regardless, this study illustrates the potential for cNLPs to deliver bioavailable curcumin to cells and serve as radiological protectants to normal tissues or as adjuvanted therapeutics for ongoing cancer therapies. Exploring higher curcumin concentrations, as well as investigating in vivo model systems, may reveal additional radioprotective qualities of cNLPs against IR-induced normal tissue toxicities, including the impact of the anti-inflammatory nature of curcumin that was not tested in this cellular model.

Overall, these data suggest that cNLPs may be useful as radiological protectants that can be applied in clinical radiotherapy to protect normal tissues while also enhancing the cell kill of actively dividing tumor tissues. It also suggests the use of cNLPs as an ideal pharmaceutical formulation to protect from chronic low doses of environmental radiological exposures, such as in deep space missions. Overall, our cNLP formulation pipeline serves as a model for forming antioxidant-loaded nanodiscs to increase the water solubility and bioavailability within in vitro human cell cultures for radiobiology studies. In the future, it will be of interest to investigate the effects of cNLPs to protect against radiological stressors in vivo or to protect against additional ionizing radiation species altogether.

## Figures and Tables

**Figure 1 nanomaterials-12-03619-f001:**
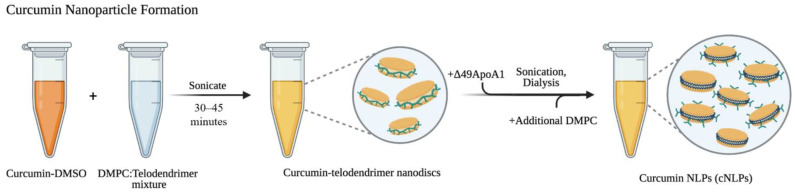
**Forming curcumin nanoparticles.** Curcumin-DMSO, DMPC, and an amphipathic telodendrimer (green) polymer are mixed to form curcumin-telodendrimer nanodiscs (cur-telodiscs). After cur-telodisc formation, Δ49ApoA1 protein (blue) is added to form curcumin-apolipoprotein nanodiscs. Removal of DMSO is achieved by dialysis overnight in 1× PBS. The next morning, additional DMPC lipid is added, and the resulting mixture contains solubilized small and large nanodiscs, collectively termed cNLPs.

**Figure 2 nanomaterials-12-03619-f002:**
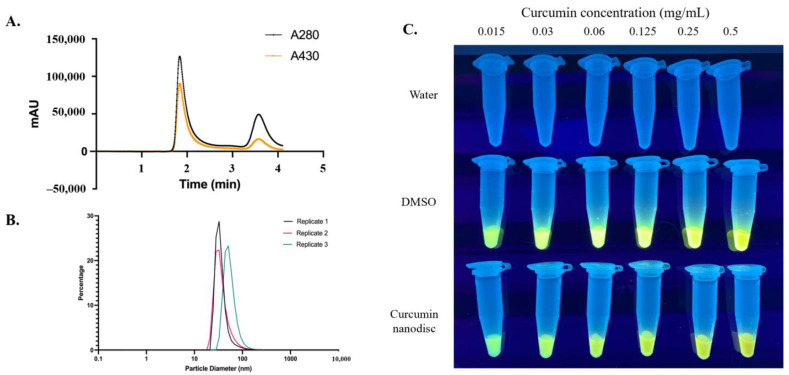
**Characterization of curcumin-NLPs (cNLPs).** (**A**) Size exclusion chromatogram demonstrates that curcumin associates within larger (leftmost peak) as well as smaller (rightmost peak) ApoA1-NLPs via co-localized 280 nm protein absorbance (black) and 430 nm curcumin absorbance (orange). (**B**) Dynamic Light Scattering of cNLPs demonstrates that these particles average about 43 nm in diameter. (**C**) Comparison of curcumin fluorescence when solubilized in water, DMSO, or curcumin-telodendrimer nanodiscs at 0.015–0.5 mg/mL concentrations.

**Figure 3 nanomaterials-12-03619-f003:**
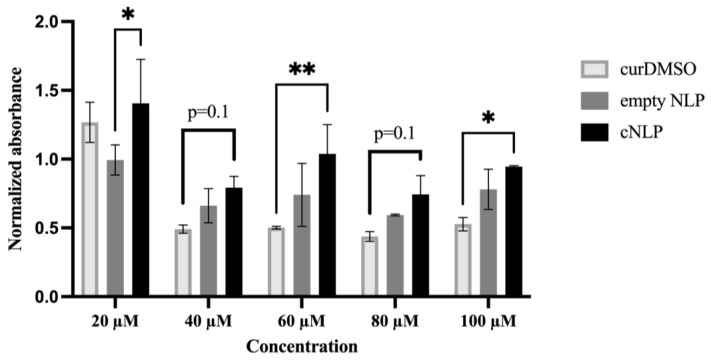
**MRC-5 cells tolerate cNLP better than DMSO solubilized curcumin.** AG05965/MRC-5 fibroblasts were pre-treated for 18 h with various concentrations of empty NLP (no curcumin) or with curcumin solubilized in nanodiscs (cNLP) or DMSO (curDMSO). MTS reagent was applied to measure cell proliferative capacity. Shown is the mean normalized absorbance ± SD. All wells were normalized to cells treated with media alone. *p*-values are labeled as <0.05 (*), <0.01 (**), <0.001 (***), and <0.0001 (****).

**Figure 4 nanomaterials-12-03619-f004:**
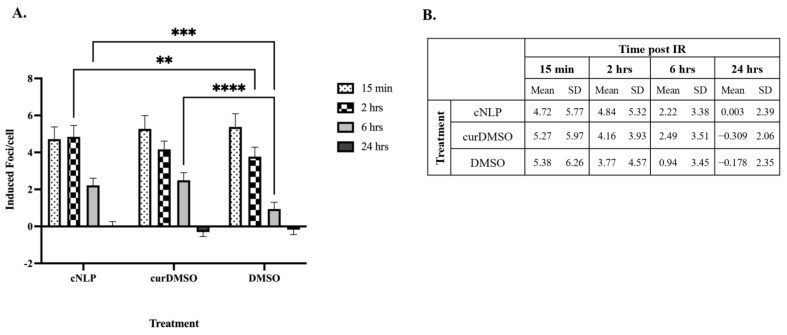
**Curcumin pre-treatment alters foci persistence following ^137^Cs****exposures.** AG05965/MRC-5 fibroblasts were pre-treated for 18 h with cNLP, curDMSO, or 0.1% DMSO alone prior to 50 cGy ^137^Cs gamma rays. Cells were fixed at 15 min, 2 h, 6 h, and 24 h post-irradiation. Data are mean induced foci per cell with 95% confidence intervals (CI, (**A**)) or SD (**B**) at each time point. *p*-values are labeled as <0.05 (*), <0.01 (**), <0.001 (***), and <0.0001 (****).

**Figure 5 nanomaterials-12-03619-f005:**
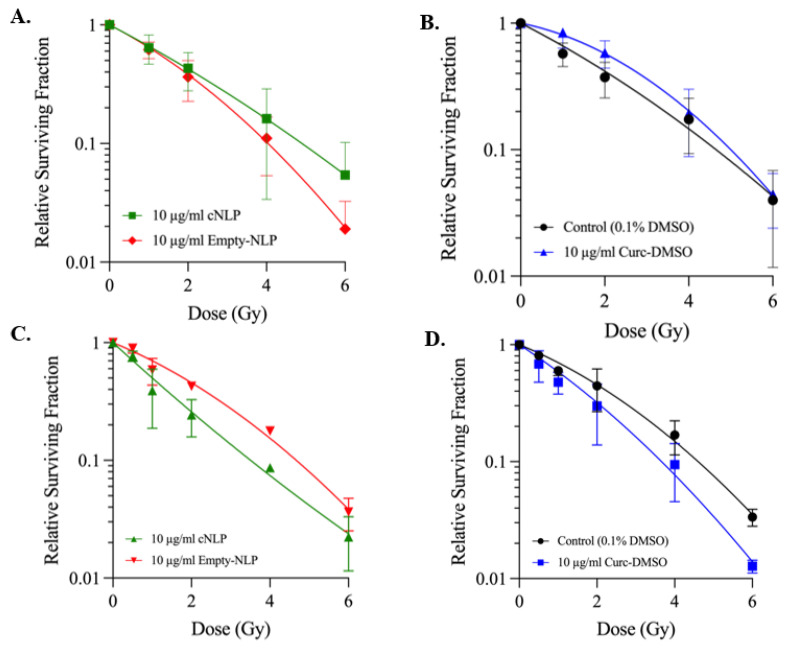
**Curcumin NLP pre-treatment alters cell survival following ^137^Cs****exposures.** Quiescent (**A**,**B**) and log-phase (**C**,**D**) AG05965/MRC-5 fibroblasts were pre-treated for 18 h with cNLP, curDMSO, empty NLP, or 0.1% DMSO alone prior to ^137^Cs gamma ray irradiation. Data are mean relative surviving fraction ± SD per treatment.

## Data Availability

Not applicable.

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
