# Peer review of "Curcumin Nanodiscs Improve Solubility and Serve as Radiological Protectants against Ionizing Radiation Exposures in a Cell-Cycle Dependent Manner"

_nanomaterials, 2022, doi:10.3390/nano12203619_

Round 1

Reviewer 1 Report

Revision:

The work entitled "Curcumin nanodiscs improve solubility and serve as radiological protectants against ionizing radiation exposures in a cell-cycle dependent manner" seems interesting to me. However, for its acceptance some questions should be resolved and the work improved.
- The introduction and material and methods are well written and give enough information.
- I would like to know why these fibroblast cells have been chosen to carry out the study.
- Figures 3 and 4 should explain the statistics better.
- In Figure 3, the ordinate axis represents the measure cell proliferative capacity?
- Figure 5 is not well explained in the text.
- The discussion should be shortened, there are many repeated concepts.

Therefore, I believe that this work needs major revision.

Reviewer 2 Report

To be honest, I am not a expert on drug delivery. Thus, based on my knowledge, several points need to be addressed carefully:

1. According to the previous literatures (since 2011), curcumin nanodiscs have been reported as carriers for drug delivery, the novelty of the research need to be specifically introduced.

2. The characterization of curcumin nanodiscs is insufficient. Would you please provide the data from fluorescence spectroscopy or SEM to prove the successful fabrication of  curcumin nanodiscs?

3.Are the concentration of curcumin  related to the diameter of curcumin incorporated nanodiscs?

4.  In vitro assay, cell up-take study is better for the demonstration of intracellular curcumin nanodics.

  •  

Round 2

Reviewer 2 Report

  • I thank the authors for careful consideration of my concerns. The authors have carefully addressed the criticisms raised by the different reviewers. I do not have additional questions. Please do spelling and gramma check for the paper. Thank you for your efforts.
